# Effect of Green Tea Extract Ingestion on Fat Oxidation during Exercise in the Menstrual Cycle: A Pilot Study

**DOI:** 10.3390/nu14193896

**Published:** 2022-09-21

**Authors:** Akira Ishikawa, Tomoka Matsuda, Hyunjun Gam, Moe Kanno, Mizuki Yamada, Nodoka Ikegami, Akiko Funaki, Hazuki Ogata, Kayoko Kamemoto, Takashi Ichihara, Mikako Sakamaki-Sunaga

**Affiliations:** 1Graduate School of Health and Sport Science, Nippon Sport Science University, Tokyo 158-8508, Japan; 2Research Fellow of Japan Science for the Promotion of Science, Tokyo 102-0083, Japan; 3Department of Exercise Physiology, Nippon Sport Science University, Tokyo 158-8508, Japan; 4Department of Judo Therapy, Teikyo University of Science, Yamanashi 405-0045, Japan; 5Waseda Institute for Sport Sciences, Waseda University, Saitama 169-8050, Japan; 6Environmental Science Research Institute, Panefri Industrial Co., Ltd., Okinawa 617-0831, Japan

**Keywords:** catechin, green tea extract, estrogen to progesterone ratio, fat oxidation, indirect calorimetry

## Abstract

In women, fat oxidation during exercise changes with the menstrual cycle. This study aimed to investigate the effect of green tea extract (GTE) ingestion on fat oxidation during exercise depending on the menstrual cycle phase. Ten women with regular menstrual cycles participated in this randomized, double-blind, crossover study. GTE or placebo was administered during the menstrual cycle’s follicular phase (FP) and luteal phase (LP). Participants cycled for 30 min at 50% maximal workload, and a respiratory gas analysis was performed. Serum estradiol, progesterone, free fatty acid, plasma noradrenaline, blood glucose, and lactate concentrations were assessed before, during, and after the exercise. Fat oxidation, carbohydrate oxidation, and the respiratory exchange ratio (RER) were calculated using respiratory gas. Fat oxidation during the exercise was significantly higher in the FP than in the LP with the placebo (*p* < 0.05) but did not differ between the phases with GTE. Carbohydrate oxidation, serum-free fatty acid, plasma noradrenaline, blood glucose, and lactate concentrations were not significantly different between the phases in either trial. Our results suggest that GTE ingestion improves the decrease in fat oxidation in the LP.

## 1. Introduction

Obesity in women has adverse health effects at each life stage [1]. Obesity has reproductive health implications associated with an increased risk of polycystic ovary syn-drome [2], gestational diabetes [3], pregnancy-related venous thromboembolism [4], miscarriage [5], and infertility [6]. In addition, fetuses born to obese women have an increased risk of inborn errors of metabolism [7]. Hence, research on obese women is required to re-duce the health risks for women and their potential future generations.

Weight loss is necessary to improve obesity, but the key is to reduce body fat mass rather than body weight. Increasing the use of fat as an energy substrate can effectively reduce body fat mass. The most effective way to achieve this is through moderate-intensity exercise where fat oxidation is approximately 10 times higher than at rest [8]. However, exercise takes time to have noticeable effects and requires effort, time, and adherence. Therefore, green tea extract (GTE), which is rich in catechins and has been proposed to easily enhance fat oxidation during exercise, has been attracting attention [9]. (-)-epigallocatechin-3-gallate (EGCG) is the most abundant catechin in green tea and is the most active catechin. Moreover, the fat-oxidizing effects of GTE, particularly EGCG, have been widely reported [10]. EGCG intake stimulates catecholamine release through the sympathetic stimulation of transient receptor potential (TRP) channels [11,12], which promotes lipolysis [13]. Previous studies have shown that acute ingestion of GTE, starting the day before the experiment, increased fat oxidation during endurance exercise in men [14] and during brisk walking in women during the follicular phase (FP) [15]. Chronic ingestion of GTE for 10 weeks [16] and 2 months [17] has also been shown to increase fat utilization during endurance exercise in men. However, a common limitation of previous studies including women is that they have not considered the menstrual cycle, which may affect fat oxidation.

The menstrual cycle involves phases of fluctuations in blood concentrations of ovarian hormones (i.e., estrogen and progesterone) that can affect energy substrate utilization [18,19]. In a study comparing the amount of fat oxidation during submaximal exercise between the FP (low estrogen and progesterone) and luteal phase (LP, high estrogen and progesterone), fat oxidation was lower in the FP than in the LP when participants exercised at 70% $\dot{V}$O_2max_ for 60 min [20], to exhaustion [21], and during 25 min of cycling at 90% lactate threshold (LT) intensity [22]. In other words, when women are trying to lose weight, GTE ingestion may improve the decrease in fat oxidation associated with the menstrual cycle and maintain high fat oxidation throughout the menstrual cycle, thereby efficiently reducing body fat mass.

Therefore, this study aimed to investigate the effect of GTE ingestion on fat oxidation during moderate-intensity exercise in the menstrual cycle. We hypothesized that fat oxidation would be reduced in the FP compared to the LP in the placebo trial (PLA), but that the difference between the phases would be eliminated in the GTE trial.

## 2. Materials and Methods

### 2.1. Participants

Sixteen healthy women with regular menstrual cycles were recruited for this study. The sample size was calculated with ANOVA repeated measures with an in-between interaction design using G*Power version 3.1.9.7. Although twelve participants were required for the calculation, the initial number of participants was set at sixteen because menstrual cycle studies are likely to exclude participants who do not have an ovulatory menstrual cycle [23]. Participants were excluded if they reported oral contraceptive (OC) use for at least six months before the experiment or a positive smoking status. Finally, of the 16 participants initially gathered, the data of 10 were analyzed (age: 21.0 ± 1.9 years, height: 158.8 ± 5.8 cm, body weight: 56.8 ± 6.9 kg, body mass index: 22.5 ± 2.0 kg/m^2^). The reasons for the exclusion of six participants were as follows: (1) irregular menstrual cycle during the experimental period (n = 1), (2) initiation of OC during the experimental period (n = 1), (3) drop out (n = 1), and (4) progesterone concentration in the LP did not meet the reference value (n = 3) (see more details in the Section 2.2). The study was approved by the ethics committee of Nippon Sport Science University (No. 019-H155, date of approval: 13 December 2019). All research was conducted in accordance with the Declaration of Helsinki. Written informed consent was obtained from all participants prior to participation in the study.

### 2.2. Menstrual Cycle Monitoring and Phase Determination

Based on the methodological recommendations regarding the influence of the menstrual cycle on the exercise response [23], the phases of the menstrual cycle were defined using the following three steps: (1) calendar-based counting, (2) urinary luteinizing hormone (LH) measurement, and (3) serum hormone analysis. First, in the calendar-based method, the participants’ menstrual cycles were recorded for at least three months prior to the start of the experiment, and if they did not have regular menstrual cycles [24], they were excluded from the experiment. Second, to accurately predict the rise in progesterone that occurs in the mid-luteal phase, an ovulation day-prediction test kit was used daily from the day the LH surge was predicted to rise until a positive test result was obtained. Finally, the participants’ ovarian hormone concentrations were measured on the day of the experiment and used to determine each phase. The reference values of serum estradiol and progesterone concentrations were set for the FP (estradiol: 19–226 pg/mL, progesterone: <0.4 ng/mL) and LP (estradiol: 78–252 pg/mL, progesterone: 5.0–21.9 ng/mL) [23].

### 2.3. Study Design

The participants came to our laboratory on five occasions. First, a preliminary test was conducted. In the subsequent four experiments, participants took either the GTE trial or placebo (PLA) trial during the FP and LP. A randomized, double-blind, crossover study design was used. Each experimental day was separated by at least one week. The order of the experiments was random, and the numbers in the first experiment were as follows: two participants were in the GTE-FP, three were in the GTE-LP, two were in the PLA-FP, and three were in the PLA-LP group. All experiments were conducted in an artificial cli-mate room with the temperature at 23 °C and the humidity at 50%.

### 2.4. Preliminary Testing

At least three weeks before the first experiment, all participants underwent an incremental test to exhaustion on a cycle ergometer (Corival 1000S; LODE B.V medical technology, Groningen, The Netherlands) to determine maximal workload capacity (W_max_). After 1 min of rest on a cycle ergometer, participants started by cycling at 75 W and increased by 25 W every 3 min. During exercise, the pedaling speed was maintained at 60 rpm according to an electronic metronome, and measurements were taken until exhaustion. Exhaustion was defined as the point at which at least two of the following were met: (1) the participant could no longer maintain the specified pedaling speed of 60 rpm, (2) the rating of perceived exertion (RPE) reached 20, (3) the respiratory exchange ratio (RER) exceeded 1.2, and (4) the participant reached the maximum heart rate estimated by age (i.e., 220—age ± 5 beats/min). Oxygen consumption ($\dot{V}$O_2_) and carbon dioxide excretion ($\dot{V}$CO_2_) were measured using the breath-by-breath method using an expired gas analyzer (AE310-S Aero monitor; Minato Medical Science, Osaka, Japan), and the average value was calculated every 10 s. RPE was recorded every 5 min using a Borg scale ranging from 6 to 20 [25]. On the day of preliminary testing, the menstrual phases were not considered because the menstrual cycle phase does not affect W_max_ [26,27].

### 2.5. Experimental Protocols

On the day of the experiment, after the participants had entered the room, body compositions (Inbody770, InBody Japan Inc., Tokyo, Japan) were measured. A flexible 22-gauge Teflon catheter (Surflor indwelling needle, Terumo, Tokyo, Japan) was inserted into the median cubital vein, and the participants were placed in the supine position and were at rest for 30 min. A three-way stopcock (Terufusion three-way stopcock with extension tube, Terumo, Tokyo, Japan) was attached to the catheter to allow repeated blood sampling during the test period. The extension tube was filled with saline solution (OTSUKA NORMAL SALINE, Otsuka Pharmaceutical, Tokyo) until blood was drawn. Expiratory gas and heart rates were continuously measured in the supine resting state for 20 min on the bed. The index of expiratory gas used to calculate resting metabolism was based on previous studies [28] and the value for 10 min before the end of measurement was adopted. The participants then underwent 30 min of fixed-load exercise on a cycle ergometer at 50% of the W_max_ determined in the preliminary testing. Blood samples were taken three times throughout the testing, pre-exercise (Pre), 15 min during exercise (Ex), and immediately after exercise (Post), and RPE was recorded at the intervals. Expiratory gases were measured continuously (Figure 1).

### 2.6. Diet

The diet on the day before each experiment was investigated using the photographic diet record method and the weighing record method. In the second and subsequent experiments, participants were asked to repeat the same diet as in the first experiment regarding the types of food, quantity, and cooking methods, while checking the dietary records. The total energy and macronutrient intakes were analyzed using nutritional analysis software (Excel Eiyokun, Kenpakusha, Tokyo, Japan). The intake of polyphenol-rich foods (e.g., green tea beverages, red wine, and high-cocoa chocolate) was restricted for 7 days prior to the experiment, and the intake of caffeine-rich foods, drugs, and alcohol was restricted for 24 h prior to the experiment. On the day of the experiment, the participants fasted for 10 h prior to the experiment and were only allowed to drink water.

### 2.7. Supplements

GTE and PLA were packed in capsules (gelatin capsules, DR T&T HEALTH, Northants, UK) and ingested in three divided doses: at lunch and dinner on the day before the experiment and 60 min before the start of the experiment on the day of the experiment. The amount of GTE (Polyphenon 70S, Mitsui Norin, Tokyo, Japan) ingested per experiment was 1200 mg (968.4 mg of total catechins including 406.8 mg of EGCG). The timing for ingestion and the amount of GTE were determined based on previous studies [14,29]. For PLA, maltodextrin (maltodextrin TK-16, Matsutani Chemical Industry, Hyogo, Japan) was used, and 1200 mg was ingested per experiment. It was confirmed that the appearance and flavor of both supplements were indistinguishable.

### 2.8. Blood Sample Analysis

Immediately after collection, 3 mL of blood was collected in ethylenediaminetetraacetic acid tubes (Neotube NP-EN0557, Nipro Corporation, Osaka, Japan) and centrifuged at 3000 rpm for 10 min at 4 °C, and the plasma was immediately frozen at −20 °C and stored until analysis. In addition, 4 mL of blood was collected in a serum separator tube (Neotube NP-SP0725, Nipro Corporation, Osaka, Japan) allowed to clot at room temperature for 30 min, centrifuged at 3000 rpm for 10 min at 4 °C, and stored and refrigerated at 5 °C until analysis. The concentrations of serum estradiol, serum progesterone, serum free fatty acids (FFA), plasma noradrenaline, blood glucose, and blood lactate were assayed. Serum estradiol and serum progesterone concentrations were measured using chemiluminescence immunoassay, serum FFA was measured using the enzymatic method, and plasma noradrenaline was measured using high-performance liquid chromatography. Blood glucose concentrations were measured using a glucose analyzer (Glucocard G Black; Arkray, Kyoto, Japan). Blood lactate concentrations were measured using a simple blood lactate analyzer (Lactate Pro2; Arkray, Kyoto, Japan). The detection range and coefficients of variation were as follows: 10–1000 pg/mL, <7% for estradiol; 0.1–40.0 ng/mL, <10% for progesterone; 0.01–4.00 mEq/L, CV < 1.5% for FFA; 0.02–9,980,000 mg/dL, <2.9% for noradrenaline; 10–600 mg/dL, <1.8% for glucose; and 0.5–25.0 mmol/L, <4.0% for lactate.

### 2.9. Substrate Oxidation

$\dot{V}$O_2_ and $\dot{V}$CO_2_ were measured using a respiratory gas analyzer, and the average values were calculated every 10 s. The obtained values were substituted into the equations of Péronnet and Massicotte [30] (Equations (1) and (2) below) to calculate the amount of fat and carbohydrate oxidation per minute of exercise. Thereafter, fat oxidation, carbohydrate oxidation, and RER were calculated as follows: the average of 10 min of rest in the supine position before exercise as Pre, the average of 1 to 15 min of exercise as Ex, and the average of 16 to 30 min of exercise as Post.
(1)$$\mathrm{CHO}\;\mathrm{oxidation}\;(g/\min)=4.585\;\times \;\dot{V}{\mathrm{CO}}_{2}\;(L/\min)-3.226\;\times \;\dot{V}O_{2}\;(L/\min)$$
(2)$$\mathrm{Fat}\;\mathrm{oxidation}\;(g/\min)=1.695\;\times \;\dot{V}O_{2}\;(L/\min)-1.701\;\times \;\dot{V}{\mathrm{CO}}_{2}\;(L/\min)$$

### 2.10. Statistical Analysis

All data obtained in this study are presented as mean ± standard deviation. To analyze the measurement items, a paired Student’s *t*-test was performed for energy and macronutrient intake. Two-way analysis of variance with repeated measures was used to compare each measurement item (serum estradiol, serum progesterone, serum FFA, plasma noradrenalin, blood glucose, blood lactate, substrate oxidation, and RER) over time (Pre, Ex, and Post). Where appropriate, multiple comparisons were made using Bonferroni’s post hoc test. Since the study aimed to identify the fat oxidative effects of GTE during each phase of the menstrual cycle, comparisons were made between FP and LP within the same supplement trials. IBM SPSS Statistics ver. 27 was used for statistical processing, and the significance level for all tests was set at *p* < 0.05.

## 3. Results

### 3.1. Menstrual Cycle and Ovarian Hormones

The average menstrual cycle of the participants was 30.7 ± 3.2 days. The number of days from the start of menstruation in the FP and LP in each trial was 3.7 ± 2.9 days and 26.1 ± 3.5 days, respectively, in the GTE trial, and 3.9 ± 2.2 days and 24.3 ± 4.0 days, respectively, in the PLA trial. The serum estradiol and progesterone concentrations are presented in Table 1. Serum estradiol concentration was higher in the LP than in the FP in both trials at all time points (menstrual cycle, *p* < 0.01, *p* < 0.01; time, *p* < 0.01, *p* < 0.01; menstrual cycle × time, *p* < 0.01, *p* < 0.01; GTE-trial, PLA-trial, respectively). Serum progesterone concentration was higher in the LP than in the FP in both trials at all time points (menstrual cycle, *p* < 0.01, *p* < 0.01; time, *p* < 0.01, *p* < 0.01; menstrual cycle × time, *p* < 0.01, *p* < 0.01; GTE-trial, PLA-trial, respectively). The E/P ratios were higher in the GTE trial (14.4 at Pre, 13.5 at Ex, and 14.2 at Post) as compared to the placebo trial (14.0 at Pre, 12.6 at Ex, and 13.2 at Post).

### 3.2. Substrate Oxidation

Substrate oxidation and RER are shown in Figure 2, Figure 3 and Figure 4. Carbohydrate oxidation during exercise was not significantly different between the phases in both trials (menstrual cycle, *p* = 0.51, *p* = 0.28; time, *p* < 0.01, *p* < 0.01; menstrual cycle × time, *p* = 0.72, *p* = 0.38; GTE trial, PLA trial, respectively). Fat oxidation and RER over the time during the exercise did not differ between the phases in the GTE trial, but in the PLA trial, fat oxidation was significantly higher in the FP than in the LP (menstrual cycle, *p* = 0.71, *p* < 0.05; time, *p* < 0.01, *p* < 0.01; menstrual cycle × time, *p* = 0.89, *p* = 0.05; GTE trial, PLA trial, respectively), and RER was significantly lower in the FP than in the LP (menstrual cycle, *p* = 0.69, *p* < 0.05; time, *p* < 0.01, *p* < 0.01; menstrual cycle × time, *p* = 0.72, *p* = 0.62; GTE trial, PLA trial, respectively).

### 3.3. Metabolites and Hormones

Table 2 shows serum FFA, plasma noradrenalin, blood glucose, and blood lactate concentrations. There were no significant differences in the concentration of metabolites and hormones between the phases in both trials (serum FFA: menstrual cycle, *p* < 0.01, *p* = 0.11; time, *p* < 0.01, *p* < 0.01; menstrual cycle × time, *p* = 0.32, *p* = 0.14; plasma noradrenalin: menstrual cycle, *p* = 0.57, *p* = 0.59; time, *p* < 0.01, *p* < 0.01; menstrual cycle × time, *p* = 0.10, *p* = 0.97; blood glucose: menstrual cycle, *p* = 0.67, *p* = 0.91; time, *p* < 0.05, *p* < 0.05; menstrual cycle × time, *p* = 0.12, *p* = 0.94; blood lactate: menstrual cycle, *p* = 0.65, *p* = 0.54; time, *p* < 0.01, *p* < 0.01; menstrual cycle × time, *p* = 0.64, *p* = 0.43; GTE trial, PLA trial, respectively).

### 3.4. RPE

Table 3 shows RPE during exercise. There were no significant differences in RPE between the phases in either trial (menstrual cycle, *p* = 0.39, *p* = 0.16; time, *p* < 0.01, *p* < 0.01; menstrual cycle × time, *p* = 0.89, *p* = 0.36; GTE trial, PLA trial, respectively).

### 3.5. Diet

Table 4 shows the energy and macronutrient intake of the participants. There were no significant differences in energy, protein, fat, and carbohydrate intake between the phases in both trials (energy: *p* = 0.20, *p* = 0.89; protein: *p* = 0.14, *p* = 0.43; fat: *p* = 0.24, *p* = 0.40; carbohydrate: *p* = 0.19, *p* = 0.27; GTE trial, PLA trial, respectively).

## 4. Discussion

The present study was conducted to compare fat metabolism during exercise after ingestion of GTE or PLA during the FP and LP in women with regular menstrual cycles. In the PLA trial, there was a significant decrease in fat oxidation and a significant increase in RER during exercise in the LP compared to the FP. In contrast, in the GTE trial, the menstrual cycle had no effect on fat oxidation. To the best of our knowledge, no previous study has investigated the relationship between changes in ovarian hormone concentrations during the menstrual cycle and the effects of GTE on fat oxidation during exercise.

As mentioned above, the decrease in fat oxidation in the LP compared to the FP observed in the PLA trial was contrary to our hypothesis, and the results of previous studies that reported that fat oxidation during moderate-intensity exercise was higher in the LP than in the FP [20,21,22,31,32]. In contrast, there have been reports of no difference in energy substrate utilization by phase of the menstrual cycle [19,33,34,35,36], and consistent results have not been obtained. This is due to inconsistencies in the methods used to validate the menstrual cycle. Many previous studies did not use serum ovarian hormone concentrations, the gold standard for defining phases, suggesting that changes in ovarian hormone concentrations during the menstrual cycle may not be properly compared [23]. This study followed the methodological recommendations [23] and defined the menstrual cycle accordingly. McNulty et al. assessed the quality of studies in their meta-analysis of the menstrual cycle and sports performance [37]. This study was identical to that with the highest quality method of categorization. Therefore, the results of this study may serve as a small step in clarifying the inconsistency of studies on the effect of the menstrual cycle on substrate oxidation. Estrogen increases in the LP and enhances fat metabolism [19,20,21,22,32]. However, there are reports of estradiol increasing glucose utilization by promoting glycolysis and citric acid circuits [38]. Since it has been shown that changes in carbohydrate oxidation are primary and may affect subsequent fat oxidation [39,40], in the present study, we speculated that increased estradiol in the LP increased carbohydrate oxidation, which was considered to cause a secondary decrease in fat oxidation.

In addition, in the LP, both estrogen and progesterone were increased [41]. Progesterone is known to have an antagonistic effect on estrogen [42,43,44]. In animal experiments, inhibition of the fat hypermetabolic effect induced by estrogen was observed when progesterone was administered in addition to estrogen [33,45]. Furthermore, when estrogen and progesterone were administered to ovariectomized rats, the activity of enzymes related to fat metabolism was significantly decreased compared to that in sham-operated rats [44]. Therefore, progesterone administration may not only inhibit the hypermetabolic effect of estrogen but may also further suppress fat metabolism. In a previous study in which ovarian hormones were administered orally to postmenopausal women, insulin sensitivity was significantly decreased in the estrogen and progesterone groups compared to the estrogen-only groups [46]. Decreased insulin sensitivity causes a decrease in fat oxidation by increasing malonyl CoA [47], which is associated with increased carbohydrate oxidation [48]. These findings suggest that the decrease in fat oxidation during the LP in this study may be influenced by an increase in progesterone. Furthermore, the E/P ratio should be considered when evaluating the effects of ovarian hormones during the LP [49,50,51]. Although no clear criteria have been established for the E/P ratio to influence energy substrate utilization during exercise, it has been reported that an increase in the E/P ratio during the LP may improve endurance performance [51]. The E/P ratio ranged from 19.0 to 31.6 when a decrease in respiratory quotient [21] and a decrease in muscle glycogen utilization during moderate-intensity exercise [20,33] were observed during the LP compared to the FP in a previous study. The E/P ratio in the PLA trials in the present study was 12.6–14.0, which was lower than those in previous studies, suggesting that progesterone, which inhibits fat metabolism, may have a stronger effect on energy substrate utilization, resulting in a significant decrease in fat oxidation during the LP compared with the FP.

EGCG, which has the highest bioactivity among the catechins contained in GTE, reaches its highest blood concentration after 60 min of oral intake [52]. Therefore, in this study, GTE was taken orally 60 min before the start of the exercise. Since the half-life of EGCG is 5 to 6 h [52], the results of the GTE trial can be considered to show the effect of GTE.

The results showed no difference in plasma noradrenaline concentrations between the phases in the GTE trial. This suggests that the menstrual cycle may not affect the lipolytic effect of EGCG ingestion via TRP channels. In addition, as in the PLA trial, ovarian hormone concentrations were significantly higher in the LP than in the FP, and no differences in fat oxidation between the phases were observed. The E/P ratio in the LP was also lower (13.5–14.4) than in previous studies, indicating a stronger expression of progesterone than estrogen, which may have resulted in the ovarian hormones acting to decrease insulin sensitivity. In contrast, GTE has been reported to improve insulin sensitivity [14]. In a report examining the effect of GTE on glucose metabolism by conducting an oral glucose tolerance test in men, insulin sensitivity was improved by 15% in the GTE trial compared to the PLA trial [14]. In an animal study, rats with decreased insulin sensitivity were fed GTE for 12 weeks, and insulin sensitivity was improved when compared to the control group [53]. Decreased insulin sensitivity in skeletal muscles may be caused by the accumulation of fat metabolites such as FFA and diacylglycerols, which inhibits insulin signaling [54]. Therefore, GTE is thought to improve insulin sensitivity by increasing fat oxidation, thereby reducing the accumulation of fat metabolites in skeletal muscles and mitigating the inhibition of insulin signaling [53]. The lack of difference in fat oxidation between the phases in the GTE trial may be the result of the antagonistic effect of GTE on insulin sensitivity caused by ovarian hormones in the LP.

This study has some limitations. First, the participants’ exercise habits and physical activity levels were not considered. To provide the same exercise load regardless of the participant’s physical condition on the day of the study, we performed a 30 min fixed-load exercise at 50% intensity of the maximal exercise load, and the physical condition was checked before the exercise. We observed that RPE increased with time, although there was no difference between the phases in the GTE and PLA trials. The increase in exercise intensity was accompanied by a decrease in fat oxidation [55]. That is, the exercise intensity in this study may have fluctuated relative to the physical condition of the participants on the day of the study and may have affected energy substrate utilization. Second, we did not measure the serum insulin concentrations. The reason for the lack of difference between the phases in the amount of fat oxidation observed in the GTE trial was the antagonism between the reduction in insulin sensitivity caused by ovarian hormones in the LP and the improvement in insulin sensitivity caused by GTE. However, this is just speculative due to the lack of actual insulin measurements. Third, the results in this study apply to young women who are healthy and have regular menstrual cycles; therefore, they may not apply to patients with lifestyle-related diseases, obesity, or those who do not have regular menstrual cycles. However, considering the necessity of obesity prevention and reduction measures for young women to reduce the incidence of obesity, GTE may be a viable option, and our study findings pave the way for further research in this regard.

## 5. Conclusions

In conclusion, our results suggest that fat oxidation during exercise is lower during the LP than during the FP, but acute GTE ingestion from lunch the day before exercise eliminates the disparity in fat oxidation caused by the menstrual cycle. Future studies are needed to determine whether chronic GTE ingestion leads to decreased body fat mass and improved fat metabolism.

## Figures and Tables

**Figure 1 nutrients-14-03896-f001:**
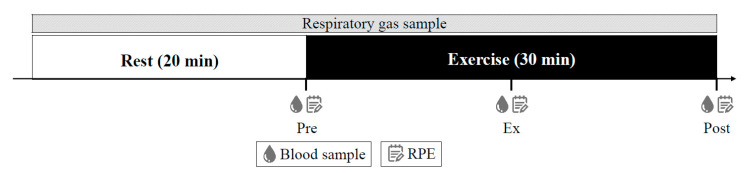
The schematic representation of the protocol. RPE, rating of perceived exertion; Pre, before exercise; Ex, 15 min during exercise; Post, immediately after exercise.

**Figure 2 nutrients-14-03896-f002:**
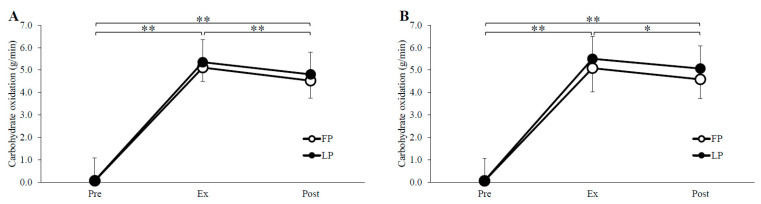
Carbohydrate oxidation for GTE (**A**) and PLA (**B**) trials. FP, follicular phase; LP, luteal phase; Pre, before exercise; Ex, 15 min during exercise; Post, immediately after exercise. * *p* < 0.05, ** *p* < 0.01.

**Figure 3 nutrients-14-03896-f003:**
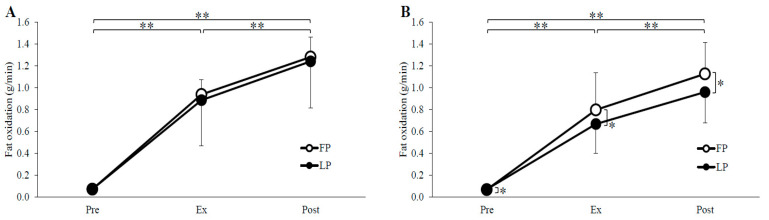
Fat oxidation for GTE (**A**) and PLA (**B**) trials. FP, follicular phase; LP, luteal phase; Pre, before exercise; Ex, 15 min during exercise; Post, immediately after exercise. * *p* < 0.05, ** *p* < 0.01.

**Figure 4 nutrients-14-03896-f004:**
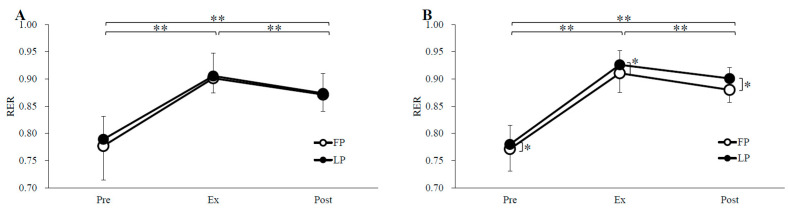
RER for GTE (**A**) and PLA (**B**) trials. FP, follicular phase; LP, luteal phase; RER, the respiratory exchange ratio; Pre, before exercise; Ex, 15 min during exercise; Post, immediately after exercise. * *p* < 0.05, ** *p* < 0.01.

**Table 1 nutrients-14-03896-t001:** Ovarian hormone concentrations before, during, and after exercise.

	Trial	Phase	Pre	Ex	Post
Estradiol (pg/mL)	GTE	FP	32.0 ± 9.4	35.4 ± 9.0	37.6 ± 11.0
		LP	119.1 ± 38.3 **	134.1 ± 39.4 **^,††^	147.9 ± 41.3 **^,††,§§^
	PLA	FP	24.4 ± 12.8	30.0 ± 15.6 ^†^	31.0 ± 17.0 ^†^
		LP	176.3 ± 55.0 **	197.3 ± 56.9 **^,†^	214.5 ± 51.1 **^,††^
Progesterone (ng/mL)	GTE	FP	0.4 ± 0.4	0.4 ± 0.4	0.4 ± 0.4
		LP	8.3 ± 3.8 **	9.9 ± 4.8 **^,†^	10.4 ± 5.1 **^,^^††^
	PLA	FP	0.3 ± 0.1	0.3 ± 0.1	0.4 ± 0.2 ^§§^
		LP	12.6 ± 3.7 **	15.7 ± 4.4 **^,††^	16.3 ± 4.8 **^,††^
E/P ratio	GTE	LP	14.4	13.5	14.2
	PLA	LP	14.0	12.6	13.2

Estradiol and progesterone were measured before, during, and after exercise in all studies. Values are mean ± SD (n = 10). E/P ratio, the ratio of estradiol to progesterone concentration; GTE, green tea extract; PLA, placebo; FP, follicular phase; LP, luteal phase; Pre, before exercise; Ex, 15 min during exercise; Post, immediately after exercise. All serum estradiol and progesterone concentrations were analyzed using a two-way analysis of variance with menstrual cycle and time as factors. ** *p <* 0.01 vs. FP; ^†^
*p <* 0.05 vs. Pre; ^††^
*p <* 0.01 vs. Pre; ^§§^
*p <* 0.01 vs. Ex.

**Table 2 nutrients-14-03896-t002:** FFA, noradrenalin, glucose, and lactate concentrations before, during, and after exercise.

	Trial	Phase	Pre	Ex	Post
FFA (μEq/L)	GTE	FP	0.56 ± 0.14	0.56 ± 0.17 ^††^	0.86 ± 0.19 ^††,§§^
		LP	0.56 ± 0.16	0.56 ± 0.27 ^††^	0.98 ± 0.52 ^††,§§^
	PLA	FP	0.54 ± 0.19	0.46 ± 0.19 ^††^	0.88 ± 0.37 ^††,§§^
		LP	0.51 ± 0.15	0.52 ± 0.33 ^††^	0.71 ± 0.19 ^††,§§^
Noradrenalin (ng/mL)	GTE	FP	0.11 ± 0.03	0.55 ± 0.15 ^††^	0.61 ± 0.18 ^††^
		LP	0.14 ± 0.04	0.63 ± 0.21 ^††^	0.57 ± 0.19 ^††^
	PLA	FP	0.13 ± 0.05	0.60 ± 0.27 ^††^	0.59 ± 0.32 ^††^
		LP	0.16 ± 0.07	0.65 ± 0.23 ^††^	0.63 ± 0.33 ^††^
Glucose (mg/dL)	GTE	FP	84.9 ± 4.5	83.2 ± 5.8	86.5 ± 6.5 ^†^
		LP	82.4 ± 8.4	86.1 ± 6.9	89.6 ± 7.8 ^§^
	PLA	FP	82.2 ± 4.3	82.9 ± 6.3	88.6 ± 6.2
		LP	82.0 ± 5.8	82.4 ± 5.2	88.8 ± 6.7
Lactate (mmol/L)	GTE	FP	1.3 ± 0.3	3.3 ± 0.9 ^††^	3.1 ± 1.1 ^††^
		LP	1.4 ± 0.2	3.6 ± 1.2 ^††^	3.1 ± 1.0 ^††^
	PLA	FP	1.4 ± 0.1	3.6 ± 1.2 ^††^	3.4 ± 1.2 ^††^
		LP	1.3 ± 0.3	3.7 ± 0.7 ^††,§§^	3.1 ± 0.7 ^††,§§^

FFA, noradrenalin, glucose, and lactate were measured before, during, and after exercise in all studies. Values are mean ± SD (n = 10). FFA, free fatty acid; GTE, green tea extract; PLA, placebo; FP, follicular phase; LP, luteal phase; Pre, before exercise; Ex, 15 min during exercise; Post, immediately after exercise. All measurements were analyzed using a two-way analysis of variance with menstrual cycle and time as factors. ^†^
*p <* 0.05 vs. Pre; ^††^
*p <* 0.01 vs. Pre; ^§^
*p <* 0.05 vs. Ex, ^§§^
*p <* 0.01 vs. Ex.

**Table 3 nutrients-14-03896-t003:** RPE concentrations before, during, and after exercise.

	Trial	Phase	Pre	Ex	Post
RPE	GTE	FP	6.6 ± 0.5	11.8 ± 1.9 ^††^	13.7 ± 2.1 ^††,§^
		LP	7.2 ± 1.0	12.2 ± 2.0 ^††^	14.0 ± 1.9 ^††,§^
	PLA	FP	6.8 ± 1.2	12.5 ± 1.7 ^††^	13.6 ± 1.3 ^††,§^
		LP	7.1 ± 1.1	12.8 ± 1.1 ^††^	14.8 ± 2.0 ^††,§^

RPE was measured before, during, and after exercise in all studies. Values are mean ± SD (n = 10). RPE, the rating of perceived exertion; GTE, green tea extract; PLA, placebo; FP, follicular phase; LP, luteal phase; Pre, before exercise; Ex, 15 min during exercise; Post, immediately after exercise. RPE was analyzed using a two-way analysis of variance with menstrual cycle and time as factors. ^††^
*p <* 0.01 vs. Pre; ^§^
*p <* 0.05 vs. Ex.

**Table 4 nutrients-14-03896-t004:** Energy and macronutrient intake of the participants.

Trial	Phase	Energy (kcal)	Protein (g)	Fat (g)	Carbohydrate (g)
GTE	FP	1281.5 ± 461.7	41.4 ± 15.5	50.9 ± 18.6	157.2 ± 59.1
	LP	1518.8 ± 592.3	50.8 ± 16.7	61.4 ± 24.5	188.9 ± 84.4
PLA	FP	1671.6 ± 633.0	57.1 ± 19.0	64.1 ± 41.0	217.1 ± 87.1
	LP	1642.8 ± 446.8	63.6 ± 25.8	75.1 ± 36.0	181.5 ± 67.3

The participants recorded a diet the day before each experiment. Values are mean ± SD (n = 10). GTE, green tea extract; PLA, placebo; FP, follicular phase; LP, luteal phase. Energy and macronutrients were analyzed using the paired Student’s *t*-test.

## Data Availability

The data presented in this study are available on request from the corresponding author. The data are not publicly available due to ethical restrictions.

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
