# Peer review of "Effect of Green Tea Extract Ingestion on Fat Oxidation during Exercise in the Menstrual Cycle: A Pilot Study"

_nutrients, 2022, doi:10.3390/nu14193896_

Round 1

Reviewer 1 Report

Ishikawa et al investigated the effect of green tea extract on fat oxidation during different phases of the menstrual cycle. This was an interesting study that shed lights on the effects of hormonal fluctuations on energy metabolism which have often been overlooked. It was great to see that the study was carefully designed to minimize/consider confounding variables. Given the complexity of the study it was perhaps understandable that the sample size was somewhat small and the outcome measures were limited. As such I wonder if the authors should consider re-writing the manuscript to position it as a pilot or proof-of-concept study, rather than drawing extensive conclusions from the limited findings. My specific comments are as follows:

Major comments:

1.    The study design section needs to be substantially revised and elaborated for clarity. What do you mean by “two to five times”? The order of randomization was also not clear to me.

2.    Was the sample size determined by power calculations? If not how was it determined?

3.    How was the dosing regimen of GTE determined?

4.    Were there any variations in the habitual intake of polyphenol-rich foods across the cohort?

Minor comments:

1.    I believe there was a typo in “1200 g” for PLA in the Supplements section.

2.    Language editing is recommended.

Author Response

Response to Reviewer #1

Thank you for taking the time to review our manuscript. We wish to express our appreciation for your insightful comments on our manuscript, they have helped us to revise and significantly improve the manuscript.

Comments

Ishikawa et al investigated the effect of green tea extract on fat oxidation during different phases of the menstrual cycle. This was an interesting study that shed lights on the effects of hormonal fluctuations on energy metabolism which have often been overlooked. It was great to see that the study was carefully designed to minimize/consider confounding variables. Given the complexity of the study it was perhaps understandable that the sample size was somewhat small and the outcome measures were limited. As such I wonder if the authors should consider re-writing the manuscript to position it as a pilot or proof-of-concept study, rather than drawing extensive conclusions from the limited findings. My specific comments are as follows:

Response

We appreciate your recognition of our research and your useful suggestions. As you have pointed out, it is very unfortunate that the sample size, which was sufficient to demonstrate detection power at the time of recruitment, was ultimately reduced. We have revised the title as a pilot study. We would appreciate your review.

Major comments 1

The study design section needs to be substantially revised and elaborated for clarity. What do you mean by “two to five times”? The order of randomization was also not clear to me.

Response

Thank you for your comment. As per your comment, we have revised the sentence. (Page 7, lines 92-96)

Major comments 2

Was the sample size determined by power calculations? If not how was it determined?

Response

Thank you for your comment. Yes, we used G*Power for power calculations. The previous study with recommendations for menstrual cycle studies (Janse DE Jonge et al., 2019) suggested that participants be recruited with the possibility of exclusion due to non-ovulatory menstrual cycle, and the present study followed suit, recruiting a larger number of subjects than calculated. (Page 5, lines 58-59)

Major comments 3

How was the dosing regimen of GTE determined?

Response

Thank you for your comment. We determined the dosing regimen based on previous studies (Venables et al., 2008; Narumi et al., 2014). The study by Venables et al. revealed ingesting GTE prior to cycling exercise at 50% Wmax intensity and the study by Narumi et al. indicated the corresponding time (Tmax) which the serum catechin concentration-time data, we used that ingestion timing and dosage as a reference. (Page 10, lines 149-150)

Major comments 4

Were there any variations in the habitual intake of polyphenol-rich foods across the cohort?

Response

Thank you for your comment. In this study, we conducted a dietary survey the day before the experiment and confirmed that there were no differences in energy and macronutrient intake between trials. Furthermore, we instructed the participants to abstain from consuming polyphenol-rich foods for one week before the experiment, so we believe we were in control on the day of the experiment. However, since long-term ingestion of GTE improves bioavailability, we believe that participants' habitual intake of polyphenol-rich foods should be investigated in future long-term interventions.

Minor comment 1

I believe there was a typo in “1200 g” for PLA in the Supplements section.

Response

Thank you for pointing this out. We have corrected it according to your comment.

Minor comment 2

Language editing is recommended.

Response

Thank you for raising this point. The previous manuscript was edited by a professional editing service. We informed the editing company of the reviewers' feedback and revised the manuscript again.

Thank you for your consideration.

Reviewer 2 Report

The work of Ishikawa et al. is interesting considering the inconsistency of studies in the literature on this topic, and particularly on the effect of the menstrual cycle on substrate oxidation.

However, major revisions are needed.

Main comments:

One of my main concerns is the small number of participants in this study: only 10 of the 16 patients enrolled were analyzed. In addition, the study has a number of limitations of which the Authors are aware (line 342). In my opinion, the Authors cannot draw valid conclusions and should state that this is a preliminary investigation both in the title and elsewhere. The conclusions should be validated with the inclusion of more volunteers in this study. I therefore propose that this study be submitted to the journal in the form of a communication, rather than as a research article. 

Minor comments:

- The paper should be properly presented with numbered paragraphs.

- The abstract should begin with a background rather than the purpose of the article. Please review the structure of the abstract carefully.

Author Response

Reply to the comments of Reviewer #2

Thank you for taking the time to review our manuscript. We wish to express our appreciation for your insightful comments on our manuscript, they have helped us to revise and significantly improve the manuscript.

Main comments

One of my main concerns is the small number of participants in this study: only 10 of the 16 patients enrolled were analyzed. In addition, the study has a number of limitations of which the Authors are aware (line 342). In my opinion, the Authors cannot draw valid conclusions and should state that this is a preliminary investigation both in the title and elsewhere. The conclusions should be validated with the inclusion of more volunteers in this study. I therefore propose that this study be submitted to the journal in the form of a communication, rather than as a research article.

Response

Thank you for your insightful suggestions. As you indicated, we recognize that the small sample size is a major concern. In the menstrual cycle study, it is recommended to estimate a large number of participants in advance to account for their exclusion from the analysis due to non-ovulatory menstrual cycles (Janse DE Jonge et al., 2019). In accordance with this recommendation, we are very disappointed that only 10 participants were finally included in the analysis of this study, even though we collected a larger number than the calculated sample size. According to your suggestions, I have added the words “a pilot study” to the title. (Page, line)

Minor comments 1

The paper should be properly presented with numbered paragraphs.

Response

Thank you for your comment. We have added the line numbers.

Minor comments 2

The abstract should begin with a background rather than the purpose of the article. Please review the structure of the abstract carefully.

Response

Thank you for your comment. We have added a sentence on the background of the study in the abstract. (Page 1, line 2)

Thank you for your consideration.